# Transplant Oncology: An Emerging Discipline of Cancer Treatment

**DOI:** 10.3390/cancers15225337

**Published:** 2023-11-09

**Authors:** Maen Abdelrahim, Abdullah Esmail, Ala Abudayyeh, Naoka Murakami, David Victor, Sudha Kodali, Yee Lee Cheah, Caroline J. Simon, Mazen Noureddin, Ashton Connor, Ashish Saharia, Linda W. Moore, Kirk Heyne, Ahmed O. Kaseb, A. Osama Gaber, Rafik Mark Ghobrial

**Affiliations:** 1Section of GI Oncology, Department of Medical Oncology, Houston Methodist Cancer Center, Houston, TX 77030, USA; aesmail@houstonmethodist.org (A.E.);; 2Cockrell Center of Advanced Therapeutics Phase I Program, Houston Methodist Research Institute, Houston, TX 77030, USA; 3Department of Medicine, Weill Cornell Medical College, New York, NY 10065, USA; 4Section of Nephrology, Division of Internal Medicine, The University of Texas MD Anderson Cancer Center, Houston, TX 77030, USA; 5Division of Renal Medicine, Brigham and Women’s Hospital, Harvard Medical School, Boston, MA 02115, USA; nmurakami1@bwh.harvard.edu; 6Sherrie and Alan Conover Center for Liver Disease and Transplantation, JC Walter Jr. Center for Transplantation, Houston Methodist Hospital, Houston, TX 77030, USA; 7Department of Gastrointestinal (GI) Medical Oncology, Division of Cancer Medicine, The University of Texas MD Anderson Cancer Center, Houston, TX 77030, USA

**Keywords:** transplant oncology, hepatocellular carcinoma, liver transplantation, cholangiocarcinoma, colorectal cancers, neuroendocrine tumor, immunotherapy, liver metastases, circulating tumor DNA

## Abstract

**Simple Summary:**

Transplant oncology is an evolving treatment ideal for patients suffering from various cancers with poor prognoses. The concept essentially is the complete removal and replacement of a diseased organ with that of a healthy donor, in order to improve the patient’s lifespan and quality of life. To attain this goal, multiple disciplines within the transplant field have converged to improve treatment plans by adjusting drug regimens and surgical techniques throughout multiple studies to increase survival results. Several of these studies have focused on hepatobiliary illnesses and therefore shown significant benefits to patient’s after receiving liver transplantation, in varying disease settings including, but not limited to hepatocellular carcinoma and colorectal cancer. As well as, expanding systematic drug therapies in different settings of cancer treatment, before curative surgery to allow a greater population to reach Milan criteria, and ultimately qualify for transplantation, and afterward in cases of disease recurrence. This article is a review of the current outlook of the transplant field for hepatobiliary cancers including treatment management, the history of emerging radical surgery, as well as the drug regimens, and other innovations that are also improving quality of life and patient survival.

**Abstract:**

Transplant oncology is an emerging concept of cancer treatment with a promising prospective outcome. The applications of oncology, transplant medicine, and surgery are the core of transplant oncology to improve patients’ survival and quality of life. The main concept of transplant oncology is to radically cure cancer by removing the diseased organ and replacing it with a healthy one, aiming to improve the survival outcomes and quality of life of cancer patients. Subsequently, it seeks to expand the treatment options and research for hepatobiliary malignancies, which have seen significantly improved survival outcomes after the implementation of liver transplantation (LT). In the case of colorectal cancer (CRC) in the transplant setting, where the liver is the most common site of metastasis of patients who are considered to have unresectable disease, initial studies have shown improved survival for LT treatment compared to palliative therapy interventions. The indications of LT for hepatobiliary malignancies have been slowly expanded over the years beyond Milan criteria in a stepwise manner. However, the outcome improvements and overall patient survival are limited to the specifics of the setting and systematic intervention options. This review aims to illustrate the representative concepts and history of transplant oncology as an emerging discipline for the management of hepatobiliary malignancies, in addition to other emerging concepts, such as the uses of immunotherapy in a peri-transplant setting as well as the use of circulating tumor DNA (ctDNA) for surveillance post-transplantation.

## 1. Introduction

Hepatocellular carcinoma (HCC) is recognized as one of the most common global incidences of cancer and poses a significant challenge in oncology care. The complex nature of the disease has established the need for a multidisciplinary approach as a crucial step toward better cancer care. Transplant oncology has recently evolved as a promising new concept for treating HCC, aiming to improve patient outcomes and quality of life [1]. So far, studies claim that liver cancer is the only solid tumor that has shown promising results with liver transplantation (LT). Thus, there are four ways in which transplant oncology can potentially contribute to the development of curative measures and axillary research in the field of hepatobiliary malignancies: investigating new concepts of treatment, which include LT; pursuing translational research in self and non-self-recognition; and linking tumor and transplant immunology. Furthermore, there is a focus on developing innovative clinical and experimental standards for accessing and utilizing the explanted liver. The use of a multidisciplinary approach to assess the field of hepatobiliary oncology can lead to identifying and overcoming the limitations of current surgical techniques [2,3]. Therefore, substantial efforts have been made to establish criteria to optimally choose HCC patients who are most likely to benefit from LT.

## 2. Concept and History of Transplant Oncology

The core concept of transplant oncology is to radically cure cancer by replacing diseased organs with healthy ones, encompassing the aim of improving the survival outcomes and quality of life of cancer patients [3]. This approach further aims to expand treatment options and research availability for hepatobiliary cancers. The evidence of this can be observed in the current management of cancers primarily dependent on a multidisciplinary approach between transplantation medicine and oncology. For example, LT has become the standard treatment for early HCC in most developed countries [4,5]. Moreover, transplant centers have witnessed a steady growth in HCC patients referred for transplantation, based on this widespread standardization [6,7,8]. In consideration of this, when the International Liver Transplantation Society (ILTS) held a scientific meeting in Rotterdam, the Netherlands, discussing the future of multidisciplinary management [9], the primary focus of the discussions was LT for HCC, cholangiocarcinoma (CCA), colorectal and neuroendocrine liver metastases, pediatric malignancy, therapies for cancer recurrence after LT, and the role of novel chemotherapeutic and biological agents to enhance transplantation outcomes. This consensus conference is considered to be a turning point in transplant oncology and resulted in the release of the first consensus recommendations and guidelines.

Research studies into genomics and cancer immunogenomics are heavily reliant on novel insights into liver cancer and are one of the crucial factors that have helped evolve the concept of transplant oncology. Due to the constant development and improvement in surgical transplantation techniques, the field of oncology has further evolved traditional resection and abridged the disparity in research and literature between tumor and transplant immunology. Moreover, sustained collaborations between applicable subspecialists, including transplant oncologists, gastroenterologists, hepatologists, interventional radiologists, transplant hepatobiliary surgeons, and immunologists, are expected to advance management and curative outcomes for hepatobiliary and other existing oncology populations [10].

### 2.1. Liver Transplantation for Hepatocellular Carcinoma

Unlike other solid organ transplants, LT is considered an exception in the field of transplant oncology since it has shown promising results in radically curing HCC. Although HCC has multiple treatment options, including chemotherapy, radiotherapy, immunotherapy, and resection, the fact that almost 90% of HCC cases occur under the setting of cirrhosis makes LT the ideal treatment option with 5-year survival rates of approximately 80% [11]. Additionally, the pathology of HCC makes treatment options like resection difficult in the later stages of the disease, so treatments viewed as comparatively less radical to LT are often infeasible. Moreover, studies have found that patients treated with LT have a lower risk of overall mortality and recurrence-free mortality than patients who undergo resection [12,13,14], though resection remains the current standard of care for patients with HCC, specifically those with existing cirrhosis and no observed portal invasions, reporting 5-year overall survival (OS) at around 50% [15]. Regardless of the benefits, there are limitations related to liver resection that have shifted research into LT as a superior curative measure, particularly related to the high rates of cancer recurrence. One study reported tumor recurrence after resection in approximately 40% of patients within the first year of treatment, creating a low disease-free survival (DFS) rate and a 10-year OS of only around 25% [14,16,17]. However, understanding the individualized nature of medical treatment means that clinicians designate specific treatment regimens for each patient under their care, and there is no singular best option for everyone. However, transplantation, with its evolved selection criteria and developing pathways to reach selection, has emerged as a better option.

Based on these and other exceptional results, LT has shown a successful ability to help improve quality of life and OS in HCC patients that qualify within Milan criteria. However, now, guidelines have been modified to help select more HCC patients who are likely to benefit from LT. Not only does eligibility criteria consider size and number, but it also considers tumor biology (including tumor markers such as alpha-fetoprotein (AFP) levels [18,19]), transplant benefit (i.e., the survival on the waitlist and after LT), and the availability of donor organs [20,21,22,23]. These modifications in eligibility criteria aim to further develop the survival outcomes of LT for HCC patients [11]. In addition, the implementation of a multidisciplinary approach involving transplant oncologists, transplant surgeons, immunologists, gastroenterologists, and interventional radiologists plays a crucial role in maximizing cancer patients’ care.

#### 2.1.1. Milan Criteria

In recent years, there have been several modifications made to the tumor–node–metastasis (TNM) classification system, and other new systems have been developed to give further insight into the ideal treatments for these patients [24,25]. Additional studies have also shown that tumor staging before transplantation is related to the rate of cancer recurrence after LT. Moreover, patients in the initial stages of HCC have shown better results with LT [26,27,28].

In 1996, Mazzaferro et al. established an eligibility model for patients diagnosed with unresectable HCC to be treated via LT, which is still considered to be the gold standard today [29]. The Milan criteria were established to determine whether HCC patients can proceed with transplantation. These criteria include a tumor diameter of a single lesion ≤ 5 cm, or for multiple lesions, ≤3 tumors, each ≤3 cm, without vascular invasion or extrahepatic metastases. Moreover, patients who met the criteria must have had their HCC diagnosis confirmed pathologically or biologically, meaning either via tissue biopsy or serum AFP assay. The results of this study demonstrated the excellent outcomes of LT indication, showing that LT could be a viable and effective treatment for HCC, especially in patients who have cirrhosis and small, unresectable HCCs. The outcomes of this study established the Milan criteria as the primary eligibility source to guide HCC patients who would ideally benefit from transplantation and paved the way for further modifications and improvement in LT survival outcomes.

#### 2.1.2. Beyond Milan Criteria

Although the Milan criteria generated excellent outcomes for post-transplant recurrence-free survival (RFS), reconciling the restriction of LT to patients with only small tumors with a high volume of patients posed a challenge. This urged research institutions and hospitals to push the boundaries of the Milan criteria and be more inclusive of patients who could benefit from LT with favorable prognoses. As a result of the effort, modifications to the Milan criteria have been explored by a multitude of transplant societies to determine whether other patients with HCC may be eligible for LT with an acceptable survival rate (5 years) after transplantation (Figure 1). Accordingly, instead of depending solely on tumor size and the number of nodules, the Milan criteria were expanded to include different tumor markers, such as AFP [30]. A handful of institutions shared in the effort to expand the Milan criteria. Examples of expanded criteria include the following: the University of California San Francisco (UCSF), Up-to-seven, Tokyo, Asan, Hangzhou, the Scientific Registry of Transplant Recipients database, Kyoto, and Kyushu (Figure 1) [31,32,33,34,35,36,37,38,39,40,41,42,43]. Currently, some criteria do not only depend on tumor size but also include the tumor markers and morphological features of HCC. Namely, French, Ontario, Edmonton, Toronto, and Metroticket 2.0 all have not only expanded criteria size beyond the Milan criteria but have also included laboratory values like AFP (Figure 1). The progressively established differential systems, starting with the French: point values for tumor size, 0, 1, 4, and AFP levels, 0, 2, 3, to correlate risk assessment. The AFP values associated with the point system, 0, 2, 3, are ≤100 ng/mL, ≤1000 ng/mL, and >1000 ng/mL, respectively. Ontario: simplifying to rely on tumor volume and AFP ˂ 1000 ng/mL, Edmonton: identical criteria with modifications in lowered tumor volume and AFP ≤ 400 ng/mL requirements, Toronto: more serious expansion obliterating Milan size and number tumor restrictions and AFP ≤ 500 ng/mL, and Metroticket 2.0: corresponding the tumors directly to AFP value (Figure 1). While UCSF, Dallas, Valencia, Up-to-seven, Kyoto, and Hangzhou have all expanded their criterion past the limits of Milan criteria, all of their selection criteria is similarly based on the tumor size and number (Figure 1). Beginning with UCSF, in 2001, sightly altering the size and number of nodules to ≤6.5 cm, ≤ 3 nodules ≤ 4.5 cm, and total ≤ 8 cm; Dallas made minimal expansions to encompass ≤ 6 cm or ≤ 4 nodules ≤ 5 cm; and Valencia followed with ≤ 3 nodules ≤ 5 cm and total ≤ 10 cm. In Italy, Up-to-seven relied heavily on the name’s sake value with the total size and number of tumors not exceeding 7, the Kyoto criteria is the only member of this group that added a biological component to its criteria but not AFP, the number of lesions ≤ 10, the diameter of lesions ≤ 5 cm, and PIVKA-II ≤ 400 mAU/ML. Finally, the Hangzhou criteria came back to size and numerical values similar to UCSF with solitary ≤ 6.5 cm, ≤ 3 nodules ≤ 4.5 cm, and total numbers ≤ 8 cm (Figure 1).

According to the corresponding studies, the outcomes of these expanded criteria are all within an acceptable range, achieving a >70% 5-year survival rate. However, to this day, the Milan criteria remains the gold standard for classifying eligible patients with HCC. Moreover, adopting neoadjuvant “downstaging” techniques has further improved outcomes as well as successfully included more patients in the Milan criteria and bridged them to liver transplantation.

For example, in a study with a total of 45 HCC patients, with random assignment to the LT group or to the control group, after being downstaged, the results showed, at data cutoff and a median follow-up of 71 months, a 5-year tumor event-free survival of 76.8% in the transplantation group versus 18.3% in the control group. In particular, a 5-year OS rate of 77.5% was reported, further supporting the fact that the downstaging of eligible HCCs beyond the Milan criteria can have excellent results with LT. Thus, post-downstaging tumor response is crucial in expanding the HCC transplantation criteria.

*A*.
*University of California San Francisco Criteria*


The UCSF, in the United States, was the first institution to take the initiative of expanding the Milan criteria. They aimed at modifying the eligibility for transplantation for HCC patients who did not initially meet the Milan criteria [44]. The expansion of the UCSF criteria includes HCC patients with a single lesion ≤6.5 cm in diameter or ≤3 lesions, ≤4.5 cm each if the total tumor diameter is ≤8 cm. This expansion of the Milan criteria resulted in an additional 5% to 20/5 benefit for HCC patients’ disease prognosis, which would have been excluded under the strict Milan criteria. Moreover, the UCSF criteria demonstrated a 72.4% survival rate compared to an 85% 4-year survival rate under the Milan criteria [38].

*B*.
*Beyond USCF Criteria*


Based on the promising survival outcomes of the UCSF criteria, transplant societies turned their efforts to maximizing the number of patients with unresectable HCC who could participate in, and wholly benefit from, these new criteria in transplant oncology. Therefore, several other studies have pursued and further expanded the UCSF criteria with progressive success in patient outcomes [45,46].

For instance, in China, researchers in Hangzhou created an eligibility framework with HCC patients whose total tumor diameter was ≤8 cm or patients with a total tumor diameter > 8 cm, pushing the criteria to not merely rely on tumor size and number. Therefore, AFP levels were also put into consideration when AFP levels ≤400 ng/mL were also included in their criteria [47]. Then, in Valencia, Spain, the expansion of these inclusion criteria to include HCC patients with ≤3 nodules, each ≤5 cm in diameter, with a total tumor diameter ≤10 cm, was implemented [48]. The results of the implementation of these criteria yielded 71% and 69% 5-year OS [49]. Meanwhile, the French criteria adopted a point scale, in which ≤2 points are indicative of minimal risk [50]. In addition, prioritization is dependent on AFP scoring, the Model for End-Stage Liver Disease (MELD) score, and time on waitlists. Only patients with HCC TNM ≥ 2 and an AFP score ≤ 2 were deemed eligible for the HCC score.

Total tumor volume (TTV) is considered the basis for the Ontario criteria. Patients’ eligibility is based on measured TTV ˂ 145 cm^3^ and AFP ˂ 1000 ng/mL [46]. In the case of the Metroticket 2.0 model, the inclusion criteria was further refined to consider both tumor size and number and actual AFP value. They determined that the total number and size of tumors (in cm) should be ≤7, and that patients should have AFP levels ˂200 ng/mL. However, if the level of AFP is 200–400 ng/mL, the criteria of the tumors will marginally change to the total amount and size of tumors being ˂5. The criteria will then further shift if the patient’s AFP levels are from 400 to 1000 ng/mL, and the total number and size of tumors should be ˂4. Generally, the mounting criteria beyond UCSF’s initial expansion have shown promising but varying 5-year OS rates. The ranges of which, on average, have been from 63% to as high as 81%. All are considered to be acceptable incremental increases to outcomes in comparison to those of standard treatment survival options without LT [45].

#### 2.1.3. Portal Vein Tumor Thrombus

In the discussion of LT as an emerging treatment option for HCC, it is prudent to observe a variant that affects a substantial portion of the HCC population and can detrimentally impact patient selection for LT treatment. Portal vein tumor thrombosis (PVVT) has an incidence rate of approximately 35–50% in HCC-diagnosed patients and corresponds to a starkly negative prognostic factor due to its pathology of increasing tumor spread throughout the host’s bloodstream, bolstering the already high risk of recurrence [51,52]. Patient outcomes with PVVT complications of HCC vary widely, depending on individual treatment response, with recent data reporting survival at ≤3 months without any intervention. However, patients under treatment have survival outcomes ranging from ≤5 months to more than 5 years, with the defining characteristic of patient longevity being tumor characteristics. Understanding the modality with which to stage HCC with PVVT based on individual tumor characteristics has only been reflected in recent staging systems, such as the Barcelona Clinic Liver Cancer (BCLC) grading system, which has negatively impacted outcomes. The BCLC system, for example, classifies all patients with vascular/portal abnormalities as having stage C HCC, which has been documented in correspondence to a sorafenib treatment regimen for downstaging [52,53]. This system’s singularity could be beneficial, considering that PVVT has been considered a contraindication for other curative measures like LT [54,55]. However, it is the later stages of PVVT that pose such problematic prognostic factors for the host with the ideal surgical technique for LT, making treatments such as the resection of damaged portal veins or transarterial chemoembolization the primary modalities for downstaging patients to curative treatment within the Milan criteria for transplantation [54,56,57]. Regardless, the curative usage of LT for those with PVVT complications of HCC is a controversial issue, and the oncological field could benefit from greater participation in the identification of standard-of-care measures to downstaging and fill the relative gap in current literature.

#### 2.1.4. Salvage Liver Transplantation

Salvage liver transplantation (SLT) was initially established as a secondary measure to liver resection in order to counteract the high rate of recurrences evoked in HCC patients. This surgical technique applies to patients who are diagnosed in the early stages of HCC and considered both resectable and transplantable, i.e., patients within the boundaries of the Milan criteria [58,59,60]. However, with recurrence rates following primary resection reported in almost 70% of cases within 5 years of first-line intervention, secondary SLT treatment or resection is evaluated based on the tumor development of HCC patients [61,62]. Lim et al. [58] published an intent-to-treat analysis of SLT and repeat hepatectomy for current HCC patients, evaluating long-term outcomes of 391 patients from 1994 to 2011 (Table 1. They found that the 5-year OS rates, calculated from patient secondary treatment, of SLT and secondary resection were equivalent at 71%. The 5-year DFS rates, calculated for the same period, showed an obvious benefit for transplantations, with SLT at 72% and second resection at 18%. Additionally, a meta-analysis by Li et al. [63] reported the long-term outcomes of SLT for 1-year survival at 82.3%; 3-year survival at 72.2%, which was equivalent to that of primary liver transplantations (PLT); and 5-year survival at 57.7% (Table 1). Their data configured for DFS were also found to be at or similar to PLT, with 1-, 3-, and 5-year outcomes at 80%, 67.8%, and 65.7%, respectively. The impressions of these studies not only show the recurrence benefits and similar survival rates of SLT comparative to PLT but also present an opportunity to improve outcomes in countries with rapidly increasing incidences of HCC and difficulties attaining liver grafts and donor organs for PLT [64].

### 2.2. Liver Transplant for Non-Hepatocellular Carcinoma Tumors

#### 2.2.1. Cholangiocarcinoma

*A*.
*Hilar Cholangiocarcinoma*


Hilar cholangiocarcinoma (HCCA) is considered one of the most challenging cancers to manage, with limited treatment options. Resection is the standard treatment; however, it has shown only a 20–40% 5-year survival in treated patients [18,19,78,79,80]. The disease also has a significant recurrence rate, and the majority of patients present with advanced disease either due to underlying parenchymal liver disease (such as primary sclerosing cholangitis) or the involvement of bilateral hilar anatomical structures [81]. Therefore, the administration of neoadjuvant therapies has shown excellent results in enhancing surgery outcomes with 5-year RFS values reported at up to 65% [81]. In addition, approaches have been utilized to boost resectability and minimize post-resection complications in preoperative biliary drainage and portal vein embolization. Moreover, it has been reported that preoperative biliary obstruction is associated with liver failure, and impaired postoperative regeneration vastly increases the risk of mortality. These indicated associated risks make biliary decompression of the future liver remnant preferred via endoscopic retrograde cholangiopancreatography or percutaneous transhepatic biliary drainage [82,83,84]. Although surgical resection is the mainstay treatment for HCCA, the extent of liver resection remains controversial despite extensive studies. On the other side of the spectrum, unresectable HCCA is another treatment challenge altogether.

Most transplant centers in the United States use the Mayo Clinic protocol of chemo-radiation followed by LT to treat unresectable HCCA. In the Mayo protocol, patient criteria were selected based on a population with unresectable CCA without extrahepatic intrahepatic metastases. Treatment for this population included irradiation plus bolus fluorouracil (5-FU), followed by brachytherapy with iridium and concomitant protracted venous infusion of 5-FU. The following maintenance period was the time allotted for supplemental chemotherapy (i.e., oral capecitabine ambulatory infusion 5-FU) until LT was performed (Table 1) [65].

In addition, Murad et al. demonstrated an RFS rate of 65% after 5 years in perihilar cholangiocarcinoma (PHC) patients who were treated with neoadjuvant followed by LT [81].

Houston Methodist institutional experience reported an excellent result for patients with locally advanced, unresectable, hilar, or intrahepatic cholangiocarcinoma (ICCA), who were treated with either the neo-adjuvant of Gemcitabine/Cisplatin with no radiation or other standard-of-care options of neo-adjuvant treatment prior to LT. This study reported that in non-Gemcitabine/Cisplatin patients, the OS was 75% at both years 1 and 2; 63% at years 3 to 5, whereas in the Gemcitabine/Cisplatin patients, the OS was 100% at both years 1 and 2; and 75% at years 3 to 5 [85,86].

Moreover, with the aim of gathering better evidence, several single-center and multi-institutional studies reported acceptable oncologic and patient survival outcomes in highly selected patients with ICCA and for those who received neoadjuvant therapy [71,87,88,89]. Evident in a recent prospective pilot study of unresectable locally advanced HCCA and ICCA, Hong et al. demonstrated excellent outcomes by adopting neoadjuvant downstaging before orthotopic liver transplantation (OLT) [89].

In a study conducted on 29 ICCA patients, the results showed that favorable outcomes after OLT can be achieved in a subgroup of patients with single ICCA tumors ≤ 2 cm or “very early” CCA [87]. According to the same study, variable factors can impact the prognosis, including tumor size, volume, microvascular invasion, and poor tumor differentiation. These findings were further corroborated in a multi-institutional international study with 48 patients who underwent OLT small ICCA [71].

In 2011, Hong et al. developed a risk stratification index to predict tumor recurrence after OLT in patients with locally advanced ICCA. Neoadjuvant radiation and systemic chemotherapy were indicated to these patients according to their score, whether low, intermediate, or high risk. The results were promising in the low- and intermediate-risk patients with locally advanced disease and acceptable tumor RFS [66,88]. The results showed that multifocality and perineural invasion, apart from the tumor size, are crucial indicators for patient RFS. This retrospective study further emphasized the potential role of neoadjuvant therapy in downstaging locally advanced HCCA and ICCA before OLT to improve RFS in the patient population (Table 1).

*B*.
*Intrahepatic Cholangiocarcinoma*


Considered to be the second most common liver malignancy, from a global perspective, ICCA tallies about 10% of all CCA cases reported. Similar to most variants of CCA, the presentation of the disease is primarily in the later stages, and only approximately 15% of diagnosed patients are labeled resectable [90]. Poor outcomes are expected with ICCA, considering resection is also considered to be the only curative outcome, for this particular aggressive cancer. Even with complete resection (R0), large population studies have shown that curative probability is about 10% and favorable 5-year survival outcomes rest, unfortunately, at 20% [91,92,93]. Alternative systematic therapies, in both neoadjuvant and adjuvant settings, have gained traction recently, with large cohort studies and promising survival outcomes. Though the efficacy of trials involving biliary cancer leaves a lot to be desired and requires more thorough study, Gemcitabine/Cisplatin has presented as the most favorable combinate treatment [94,95,96]. Moreover, with the additional study interest in downstaging aggressive malignancies to LT, further trials will assist in improving outcomes and efficacy data.

Lunsford et al. reported a prospective case series of patients who received neoadjuvant intervention for unresectable locally advanced ICCA and achieved stable conditions, eventually progressing to OLT [97]. The established patient inclusion criteria were tumors the size of >2 cm and multifocal disease without vascular or lymph node involvement. Based on the protocol criteria, an established minimum of 6 months of radiographic response or stability was required before the patient was allowed to progress to OLT. The results showed a 5-year OS rate of 83.3% and a 5-year RFS rate of 50% [98].

Several transplant institutions demonstrated poor results in ICCA with OS rates up to 40% at 3 years and 20% at 5 years after LT, making ICCA patients ineligible for LT (Table 1) [68,69,99].

However, researchers at the University of California succeeded in developing a prognostic scoring system to improve surgery outcomes [88,100]. Their recommendation is to use neoadjuvant/adjuvant chemotherapy, such as 5FU- or capecitabine-based regimens in combination with oxaliplatin, leucovorin calcium, and gemcitabine hydrochloride. To further optimize the results, they also suggested thorough studying of tumor biology prior to neoadjuvant therapy to further optimize the results. This method also specifically recommends evaluating tumor pathological status by obtaining tissue biopsy prior to neoadjuvant therapy initiation, further initiating the utilization of a criterion of biological factors [66]. The scoring system considers seven clinicopathological risk factors: perineural invasion, infiltrative subtype, lack of neoadjuvant or adjuvant treatment, multifocal tumor, HCCA, history of primary sclerosing cholangitis, and lymph vascular invasion. This scoring system ranks patients’ risk for recurrence in classification groups of low, intermediate, and high to select candidates for LT [88]. The patients in the low-risk group had a 78% 5-year RFS rate in comparison to those in the intermediate-risk group who were at 19% and 0% for the high-risk group [88].

Houston Methodist J.C. Walter Jr. Liver Transplant Center and MD Anderson Cancer Center had the first multi-site collaboration that published a prospective case series of patients with ICCA treated with protocolized neoadjuvant chemotherapy abridged to LT [98]. The reported series used no specific tumor size cutoff. Although, the median cumulative tumor diameter for the participating patient population was 14.2 cm. The six patients involved in treatment concluded with a 5-year OS of 83.3% and a 50% RFS [70]. Granted that cirrhosis in ICCA patients was a contraindication for LT in most transplant centers, some studies showed that “very early” ICCA may have acceptable results after LT (Table 1) [71].

#### 2.2.2. Hepatoblastoma

Hepatoblastoma (HBL) has been reported as the most common primary hepatic malignant neoplasm diagnosis in childhood alongside HCC. Historically, treatment was attained via the complete resection of malignant tumors, and while that remains the standard today, chemotherapeutic regimens have revolutionized the system by which patients qualify for curative resection [101,102]. However, the consideration of the patient is most important when determining treatment modalities; importantly, most HBL patients are diagnosed before the age of 5, and prolonged chemotherapy treatment to reach tumor resectability should be avoided [103]. For the cases of patients with more extensive tumors, studies have demonstrated children’s response to LT with chemotherapy combinates has shown to have superior outcomes in providing long-term DFS for those diagnosed with advanced-stage HBL and HCC. The staging of the disease is presented differently in children than in adults, as shown in a study of Pham TA et al. [72], who divided patients into standard- and high-risk groups. This study demonstrated comparative outcome data in which the pretreatment extent of disease (PRETEXT) stage IV tumors were significantly linked to recurrence and death in malignancies, opposing the relative, but not absolute, contraindication to transplantation in cases of metastatic HBL, which the study claimed to be a curative option (Table 1). Furthermore, it has been established that the more time a patient spends on the transplant waiting list, the greater the associated risk for the recurrence of HBL. Although HCC in children is rare, it is considered especially difficult to treat because it behaves more adversely than in adults. The criteria for evaluating transplants are different: instead of using the Milan criteria as the standard for lesions, in children under 18, the criteria is well outside both the Milan and UCSF criteria. In addition, it is resistant to chemotherapy, which makes complete resection the only available treatment. Therefore, further studies are required to establish the safe and effective role of transplants in children under 18 with HCC.

## 3. Liver Metastases

### 3.1. Neuroendocrine Tumor Liver Metastases (NETLM)

Despite the high recurrence rates following resection, surgical treatments remain among the most beneficial approaches for treating patients with NETLMs. However, to improve the survival rates following surgical treatment for NETLM, it is recommended to include resection and cytoreductive surgery [104].

According to the results of 44 cases of resection, Foster and Berman remarked good symptom control was achieved in a majority of the patients observed with at least 95% debulking as well as non-rapid rates of tumor growth [105,106]. McEntee et al. reported a resection study of 37 patients who underwent the procedure for the purpose of symptom relief. The results of the study considered symptom control to be notably achieved only if ≥90% of grossly visible tumors were successfully resected, and no specific debulking threshold was established [107]. Further studies conducted at the Mayo Clinic also supported the previous evidence. Based on the results of 74 patients who underwent resection, a debulking threshold of 90% was set. Additionally, a mean duration of response of 19.3 months, with a 4-year survival rate of 73%, and a postoperative symptomatic response rate of 90% were reported [108]. These studies set a threshold for curative surgical measures, such as resection to be further enacted on patients to expand surgical techniques and improve overall patient survival for early-stage disease. Beyond this, for patients with locally advanced, unresectable NETLM who underwent treatment for LT, recent data indicate 5-year OS ranging from 50 to 70%. According to the same data reported within a review by Morris et al., NETLM patients were also reported to have had recurrence rates from 30 to 60% over a 5-year period [109]. Mazzaferro et al. [110] established criteria with the aim of improving the results of surgery (Table 1). These criteria included patients with a low-grade NET as the primary tumor, drained via a portal system, with at least 50% hepatic involvement, who reported response to therapy or had stable disease for at least 6 months and were ≤55 years of age. The success of the Milan criteria for NETLM was demonstrated in the study results, which yielded a 90% 5-year survival rate and 89% 10-year survival rate in 42 patients, including patients who received LT between 1995 and 2010 [110]. Accordingly, new guidelines were adopted based on the Mazzaferro criteria for including patients with unresectable NETLM in patients potentially eligible for LT [62,111].

A secondary surgical option, utilized for nonlocalized tumor invasions, is multivisceral transplantation (MVT), or multiorgan transplantation, another curative treatment that involves potentially taking multiple abdominal organs and part of the lymphatic system out of the body to irradicate carcinoma. Though MVT has the potential to achieve better curative resection of metastasized tumors in the abdominal cavity, the lack of direct access to MVT centers prevents the technique from becoming a standard therapy option [112]. The major reason it has been presented as a more comprehensive treatment measure, beyond its radical methods, is the possibility of metastases in portal drainage and the lymphatic system that would otherwise be missed in a primary LT [109,113]. Morris et al. [109] published a systematic review of LT and MVT specific to NET invasions, which remains one of the only reports comparing post-LT outcomes between the two surgery techniques (Table 1). The authors found that only 16 in 279 (5.7%) transplantable patients experienced MVT for NETLM and identified that, of the 28 transplant centers in the US, only 17 MVTs occurred from 1988 to 2012 [114]. Even if other study data may show MVT to have a better curative outcome, the lack of accessibility is going to affect outcomes as much as the lack of existing literature. Further studies need to be conducted to establish standard therapy and care options for better outcomes in NETLM.

### 3.2. Colorectal Cancer

Colorectal cancer (CRC) is the third most common cancer worldwide, and fourth in terms of mortality. Metastatic variations in the disease are most commonly found in the liver and tend to affect males at a higher global incidence [115,116]. Treatment options for CRC patients with affected organs like the liver have good survival outcomes, as reported with curative hepatectomy used to treat liver oligometastases. However, often, surgical resection for hepatic ailments, dependent on the disease criteria, is explored as an option. In colorectal liver metastasis (CRLM), the treatment options include R0 resection, which is the resection process of sparing at least two adjacent liver segments having independent inflow, outflow, and biliary drainage. The remaining liver, following resection, should not be less than 20–30% of the total natural liver volume in normal and cirrhotic patients. In the presence of CRC patients with unresectable liver metastases, the initial experience of LT was not encouraging, with a 5-year OS rate lower than 20% [111]. The general consensus to the discussion of poor outcomes in patient cohorts are attributed to the absence of suitable selection criteria and the lack of appropriate neoadjuvant and adjuvant therapies. However, more studies have been conducted for better evidence (Figure 2) [117]. In the past, beginning with SECA 1, in 2008, the criteria of pre-transplant tumor diameter at > 5.5 cm and the potential for high hepatic tumor load, as well as carcinoembryonic antigen (CEA) levels before LT at >80 ug/L. A secondary study and expansion of data and results took place in the SECA 2 study in 2011, a multi-arm trial, which created the standardized Oslo score for colorectal liver metastasis patients. Later studies (Figure 2) developed the CRLM criteria by progressively building from aspects of precedence, with SECA 3, COLT, SOULMATE, MELODIC, and EXACALIBUR1 reporting comparative trials following LT versus chemotherapy, standard of care, or best alternative therapy. A recent publication from the US in 2022, a single-arm trial using living donor liver transplantation (LDLT) with the necessary criteria including computed tomography (CT), showed stable or partial response for 3 months, unresectable diagnosis, and no evidence of extrahepatic disease (Figure 2). Additional studies have also been conducted to primarily improve outcomes and evaluate optimal dosing and downstaging for CRLM patients.

For example, a study conducted by Adam et al. [118] showcased a 1104-case series of patients with an initially unresectable liver metastasis. The results of the report showed 33% 5-year survival, following primary CT, compared to 12% of patients who were resected; this value is approaching the 5-year survival rate of resectable patients in the same period, which was equal to 48%. Other studies evaluating chemotherapeutic regimens have demonstrated that patients can be downstaged from unresectable to resectable. However, the variation in patients eligible for downstaging ranges widely, with study data indicating anywhere from 15 to 50% of patients. Moreover, the optimal downstaging regimen is still an open debate, especially considering the optimal time of resection is another matter of dispute among publishing authors. Among them, some investigators claim that resection is necessary to be performed as soon as the operation is feasible for the patient’s individual lesions. Whereas others side with the argument that resection operations should only occur in the two instances when maximum response is possible (usually 4 months) and at first subsequent progression, which is usually 9 months [119]. Clavien et al. [120] published data containing a conversion rate of nearly 30% for regional liver arterial infusion (HAI) floxuridine (FUDR), which directly conflicts with a study by Kemeny et al. [121], in which they observed a rate of conversion >50% for an intervention regimen combining HAI FUDR with systemic FOLFOX.

Further demonstrating the global incidence of these diseases, in Europe, Hagness et al. [77] reported on a pilot study of long-term OS following LT for patients with CRLM (Table 1). This specific cohort was an unresectable patient population with traditionally poor prognoses, but the results of this prospective pilot study showed good outcomes, with a 5-year OS rate at 60%, and any reported recurrences were accessible for resection. Additionally, based on these findings, axillary clinical trials have demonstrated response rates exceeding 50% in unresectable liver metastatic lesions, with varying rates of 43–81% published when the molecularly targeted drug bevacizumab or the anti-epidermal growth factor receptor (EGFR) antibodies cetuximab or panitumumab are added to the study interventions [122].

## 4. Emerging Concepts in Transplant Oncology

### 4.1. Immune Therapy in the Peri-Transplant Period

#### 4.1.1. Pretransplant Bridging Therapy

Although LT in HCC shows promising results, it is only applicable to a small ratio of patients who meet the standards of the Milan criteria. Therefore, neoadjuvant therapies may be useful for downstaging tumors and hindering their progression [123,124,125]. Immune checkpoint inhibitors (ICPIs) have demonstrated significant success in improving outcomes and evolving treatment regimens for a wide range of afflicted patients (Table 2) [126]. Immune checkpoint proteins include cytotoxic T-lymphocyte-associated-4 (CTLA-4) and programmed cell death protein 1 (PD-1), which are the receptors expressed on the surface of cytotoxic T cells. These receptors work by downregulating T-cell activation to sustain peripheral tolerance as well as helping cancer cells to escape from cytotoxic T-cell-mediated death [127]. Various studies have evaluated and demonstrated the potential antitumor activities and acceptable safety profiles of ICPIs in HCC treatment. However, the results have shown successful ICPI usage across different oncology populations, and existing apprehensions about postoperative fatal rejection have perpetuated an environment where they are seldom included in the treatment of patients receiving solid organ transplants [128]. Recent research at Houston Methodist has been exploring the clinical factors that could play a significant role in rejection rates, evaluating the period between ICPI and LT called the “wash-out” period. This period is a gap between systematic treatments and transplantation that allows the regulation of the host immune system to “wash-out” the PD-1 and CTLA-4 binding receptors. It is the blocking of immune, B7, pathways that may cause these T cells to become more active, resulting in T-cell-mediated graft rejection.

Several ICPIs, such as a monotherapy of nivolumab and pembrolizumab, and in combination, such as nivolumab plus ipilimumab, or in combination with other U.S. Food-and-Drug-Administration-approved therapies, such as atezolizumab plus bevacizumab (VEGF inhibitor), have shown a significant improvement in survival outcomes and overall response in patients with unresectable HCC. Results have concluded that ICPIs can be well tolerated, despite studies documenting a wide range of adverse events (AEs), with only 15% of patients classified as unresectable HCC suffering from AEs that require any treatment discontinuation. However, ejection and graft loss still pose unmet challenges [129]. Although the use of ICPIs is rapidly evolving in the field, the safety of ICPI therapy remains questionable and requires further investigation. A study recently reported that nine patients with HCC were transplanted after receiving nivolumab as a neoadjuvant intervention at a single center [130], and 16 months after receiving transplantation, at their median follow-up, there were no reported severe allograft rejections/losses. Additionally, over the same median follow-up period, there were no reported tumor recurrences or deaths. However, a single patient had developed mild acute rejection due to low tacrolimus levels; however, after immunosuppressant levels were corrected, the issue resolved itself soon after. In the explant liver, about a third of evaluated patients had near complete (>90%) tumor necrosis [130]. Despite the promising results, this report concluded that further prospective studies of ICPIs in the pretransplant setting are required for a better understanding of the optimal interventive utilization of ICPIs in patients waiting for LT.

Transarterial chemoembolization (TACE) is another downstaging technique that showed promising results in the early days of HCC treatment and has now become a standard-of-care intervention with chemotherapy and immunotherapy combinations in several studies. Monden et al. [131] reported on one of the earliest experiences of TACE in a clinical setting. A total of 71 patients treated preoperatively with TACE were compared to 21 patients resected without TACE. Although the study did not determine that there were any significant differences in survival, a histopathologic review concluded that there were signs of tumor necrosis in patients who underwent TACE preoperatively. In another retrospective study, Zhang et al. [132] studied 1457 HCC patients who underwent hepatic resection, including 120 patients treated preoperatively with TACE, and compared the results to those resected without TACE. The evaluation revealed that patients who underwent preoperative TACE had significantly improved 5-year DFS. Additionally, patients documented to have had more than two preoperative TACE treatments showed longer RFS compared to those who only had one session. Over a 10-year period, Zhang et al. [55] also showed that from 831 patients treated with TACE, 82 patients were successfully downstaged, and 43 subjects underwent salvage surgery. Patients who underwent resection had a longer median OS (49 months vs. 31 months, *p* = 0.027) when compared to those who refused a salvage resection. However, the results showed no significant difference in survival outcomes based on those who received surgery and experienced a complete response (CR) (50 months vs. 54 months, *p* = 0.699) versus those with a partial response (PR) (49 months vs. 24 months, *p* < 0.001). Findings such as these suggest that the role of resection is critical, following downstaging with TACE, in patients with PR. However, in some other studies, TACE did not improve DFS or OS nor were there any differences in 1-, 3-, and 5-year OS, and there was an increase in hospital costs associated with the procedure [133]. In conclusion, further investigation is needed to determine if TACE can positively impact LT outcomes.
cancers-15-05337-t002_Table 2Table 2Summation of the utilization of ICPIs in thirteen case reports as neoadjuvant therapy in a pre-LT setting for HCC patients.Age/SexICPI AgentICPI CycleICPI ClassInterval Time from Last Dose of ICPIs to TransplantISTType of ResponseGraft OutcomeReferences66 MAtezolizumabBevacizumab(6)(5)PD-L1VEGF60 daysTacrolimus/MMFRNo rejectionAbdelrahim et al. [134]64 MNivolumab(23)PD-116 daysMMF/Prednisone/tacrolimusRResolved rejectionAby et al. [135]39 MToripalimabLenvatinib10UKPD-1TK93 daysTacrolimus/MethylprednisoloneDGraft rejectionChen, G.H. et al. [136]64 MNivolumab(1)PD-17 daysTacrolimus/MMFRCNo rejectionChen, Z. et al. [137]47 FNivolumab(1)PD-1122 daysTacrolimus/MMFRCNo rejectionChen, Z. et al. [137]50 MNivolumab(1)PD-162 daysTacrolimus/MMFRNo rejectionChen, Z. et al. [137]38 MNivolumab(6)PD-159 daysTacrolimus/MMFRNo rejectionChen, Z. et al. [137]67 MNivolumab(6)PD-167 daysTacrolimus/MMFRNo rejectionChen, Z. et al. [137]60 MNivolumab(17)PD-15 weeksTacrolimus/MMF/steroidRGraft rejectionDehghan et al. [138]14 MPembrolizumab(3)PD-1138 daysSirolimus/tacrolimusRNo rejectionKang et al. [139]63 MNivolumabIpilimumabUKPD-1CTLA-49 weeksMethylprednisolone/ThymoglobulinRNo rejectionLizaola et al. [140]65 MNivolumabUKPD-18 daysTacrolimus/MMF/PrednisoneDGraft rejectionNordness et al. [141]68 MNivolumabUKPD-110 monthsUKRNo rejectionPeterson et al. [142]**ICPI:** immune checkpoint inhibitor, **M:** male, **F:** female, **PD-1:** programmed death, **mg:** milligram, **D:** death, **MMF:** Mycophenolate mofetil, **UK:** unknown, **IST:** immunosuppressive therapy, **PD:** a progressive disease, **R:** response, **RC:** recurrence, **OF:** organ failure, **TK:** tyrosine kinase, **CTLA-4:** cytotoxic T-lymphocyte-associated antigen 4.


#### 4.1.2. Post-Transplant Palliative Therapy

Immune therapy in the post-transplant setting has been thought to be a contraindicate in solid organ transplant recipients due to safety issues, meaning those patients will have a higher risk of allograft rejection. Although several published cases have reported some LT recipients may be treated with ICPIs in an appropriate, and differentiated, setting (Table 3). Other reports of LT recipients treated with ICPIs have portrayed a nearly two-thirds majority allograft preservation in patients [143,144], where the disease control rate of the cohort was reported at 21% and total graft rejection was seen in 37% of LT subjects. Trepidations toward recommending transplantation, on the side of the clinician, in this setting revolves primarily around the pressure on the host’s already compromised immune system, causing an enormous shift in order for the body to adjust to a new foreign entity. Then, initializing a regimen of ICPI, which is essentially meant to induce an immune response in patients, may cause the body to attack the LT. Though multiple studies have now been conducted in the palliative setting with immunosuppressants and ICPIs accompanied by careful dose management and observation to prevent graft rejection.

Munker and DeToni reviewed publications on 14 confirmed cases of LT recipients who had undergone treatment consecutively with ICPI [128]. The authors concluded that organ susceptibility to rejection depended primarily on three components of treatment: the agent of immunosuppression utilized, the status of PDL-1 in liver graft biopsies, and the time of treatment initiation. In accordance with this report, only 4 out of the 14 cases evaluated (28%) reported liver graft rejection, with the median time of rejection occurring within 3 weeks of immune therapy initiation. Survival outcomes were available in 12 of the cases reviewed, with a median value of 1.2 months in this study. Furthermore, Rammohan et al. [144] reported on an HCC occurrence case that appeared in the lung 3 years after initial living donor LT treatment. After an initial failure to respond to sorafenib, the patient was prescribed additional cycles and showed a dramatic response to the ICPI pembrolizumab, which was administered at 200 mg for 21 days along with sorafenib. After 10 months on the scheduled ICPI regimen and sorafenib, the patient remained stable and had no observed or radiological evidence of tumor or graft rejection/dysfunction [144]. De Bruyn et al. reported 19 LT patients treated with ICPIs for advanced malignancies; following this study, 21% of reported patients showed disease control and ˂38% of them reported graft rejection. However, this series is only one example in which the conclusion suggests that LT recipients can be successfully treated with ICPIs [143]. In another retrospective study, Abdel-Wahab et al. [145] evaluated 39 patients with allograft transplantation and observed 11 of 39 patients (28%) progressing to LT. The median time for this study, between ICPI initiation, for ICPIs including both anti-PD-1 and anti-CTLA-4 therapy, was 9 years post-LT. Additionally, of the enrolled hepatic patients only 4 of 11 experienced allograft rejection (41%). Although data from a singular report cluster is inadequate to obtain direct and conclusive evidence that a specific ICPI or immunosuppressant agent has greater efficacy than another, various protocols were suggested to determine these factors, such as that liver allografts tissue should be biopsied routinely before any treatment initiation in LT recipients, pre-treatment with immunosuppressants should be tried in the absence of contraindications, and immunosuppression should be tapered progressively under close surveillance. Moreover, the following laboratory parameters should be assessed: complete blood count, comprehensive metabolic panel (including kidney, liver, pancreatic, and thyroid function tests), and baseline oxygen saturation (including a “walking oxygen saturation” test to facilitate the detection of a decrease in oxygen saturation levels that might warrant further diagnostic imaging).

Currently, the use of ICPIs as a treatment possibility in the palliative setting post-LT is still under investigation. This can be primarily attributed to the gap in the number of viable cases to evaluate and coupled with insufficient literature about the relationship between graft rejection and tumor response. However, there could also be correlative clinical factors that may increase the rejection rate in a similar fashion. There are also boundaries that stagnate the progression of research, such as the limited number of predictive biomarkers that can be adapted to HCC patients undergoing immunotherapy in the post-LT, palliative setting [60]. However, the utilization of immunotherapy in the neoadjuvant setting of transplant oncology has shown promising outcomes and earned greater acceptability among the community of transplant oncology. More prospective data will be needed, in the future, to uphold its safety and efficacy.
cancers-15-05337-t003_Table 3Table 3A summation of 13 case reports on the utilization of ICPIs as palliative therapy in the post-LT setting for HCC patients.Age/SexICPI AgentICPI CycleICPI ClassInterval Time from Transplant to ICPIsISTType of ResponseGraft OutcomeReferences70 MNivolumab(4)PD-133 monthsTacrolimus/high-dose steroids.PDNo rejectionAl Jarroudi et al. [146]62 FNivolumab(5)PD-112 monthsTacrolimusPDNo rejectionAl Jarroudi et al. [146]66 MNivolumab(6)PD-124 monthsTacrolimusPDNo rejectionAl Jarroudi et al. [146]56 MNivolumab(6)PD-132 monthsTacrolimusPDNo rejectionDeLeon et al. [147]55 MNivolumab(5)PD-194 monthsSirolimus/MMFPDNo rejectionDeLeon et al. [147]34 FNivolumabUKPD-144 monthsTacrolimusPDNo rejectionDeLeon et al. [147]63 MNivolumabUKPD-114 monthsTacrolimusUKNo rejectionDeLeon et al. [147]68 MNivolumabUKPD-113 monthsSirolimusUKGraft rejectionDeLeon et al. [147]41 MNivolumab(15)PD-116 monthsTacrolimusPDNo rejectionDe Toni and Gerbes et al. [148]70 MPembrolizumab
PD-196 monthsLow-dose (50%) TacrolimusPDNo rejectionVarkaris et al. [149]53 FNivolumab(1)PD-136 monthsEverolimus/MMFD due to OF(2 weeks after start ICPI)Graft rejectionGassmann et al. [150]14 MNivolumab(1)PD-136 monthsTacrolimusD due to OF(5 weeks after start ICPI)Graft rejectionFriend et al. [151]20 MNivolumab(2)PD-148 monthsSirolimusD due to OF(4 weeks after start ICPI)Graft rejectionFriend et al. [151]61 MNivolumab(2)PD-124 monthsUKRGraft rejectionGomez et al. [152]57 MPembrolizumab(13)PD-136 monthsTacrolimus/MMF/steroidRNo rejectionRohmann et al. [144]64 MNivolumabLess than (1)PD-124 monthsThymoglobulinRGraft rejectionKumar et al. [153] 54 FIpilimumab(17)CTLA-484 monthsTacrolimus/EverolimusPRNo rejectionPandey et al. [154]54 MCamrelizumab(13)PD-148 monthsTacrolimusPDNo rejectionQui et al. [155]54 MNivolumab(12)PD-124 monthsTacrolimusPDNo rejectionZhuang et al. [156]46 MToripalimab(6)PD-112 monthsSirolimusPDNo rejectionShi Gm et al. [157]35 MAtezolizumab(12)PD-L148 monthsUKPDNo rejectionBen Khaled et al. [158]35 MPembrolizumab(2)PD-1240 monthsMMF/SteroidRNo rejectionSchvartsman et al. [159]54 MNivolumab(3)PD-1156 monthsTacrolimus/Everolimus/PrednisonePDNo rejectionBiondani P et al. [160]62 FIpilimumabPembrolizumab(4)(25)CTLA-4PD-114 monthsSirolimus/MMFPRNo rejectionKuo JC et al. [161]52 MNivolumab(4)PD-132 monthsCyclosporine/MMFPDNo rejectionKondo et al. [162]72 MNivolumab(2)PD-184 monthsMMF/BudesonideUKNo rejectionDeylon J et al. [163]59 MToripalimab(8)PD-116 monthsSirolimusPDNo rejectionShi GM et al. [157]**ICPI:** immune checkpoint inhibitor, **M:** male, **F:** female, **PD-1:** programmed death, **mg:** milligram, **D:** death, **MMF:** Mycophenolate mofetil, **UK:** unknown, **IST:** immunosuppressive therapy, **PD:** progressive disease, **PR**: partial response, **R:** response, **RC:** recurrence, **OF:** organ failure, **TK:** tyrosine kinase, **CTLA-4:** cytotoxic T-lymphocyte-associated antigen 4.


### 4.2. Utility of Circulating Tumor DNA (ctDNA) for Cancer Minimal Residual Disease (MRD) Evaluation and Surveillance

Minimal residual disease (MRD) has several established strategies of surveillance in HCC patients, such as radiological imaging and tissue biopsy. Axillary avenues of development, such as liquid biopsy, used to assess ctDNA show a favorable ability in MRD surveillance for primary liver malignancies [164,165,166]. When the molecular fragments derived from the HCC malignancy are excreted into patient bloodstreams and need to be measured and analyzed, ctDNA biopsy is currently the tool being utilized. Strategies behind the ctDNA biopsy primarily offer a noninvasive approach but also offer a resolution to the limited access that remains to HCC tissue samples obtained via standard tissue biopsy. In addition, a ctDNA biopsy reveals an entirely novel and dynamic image of HCC, a process that can be reproduced as necessary, and provides real-time surveillance for MRD in HCC patients. Any associated cost-saving benefits can be considered an added bonus. There are several studies utilizing ctDNA and MRD surveillance to demonstrate their usefulness in the clinical treatment of HCC patients [167]. Kasi et al., for example, analyzed 200 plasma samples from 90 hepatobiliary patients, and in these sample patients, they were able to identify that 27 had HCC [79]. After the study conclusion, it was reported that the detection of ctDNA should be significantly associated with the stage of disease in which it is observed. In addition, serial time point analyses have been conducted on a subset of 56 patients who had 2–7 set time points available. According to these analyses, correlations between the clinical response and ctDNA levels were demonstrated to an appropriate degree [168,169].

Furthermore, the clinical uses of ctDNA biopsy for the detection of tumor progression, MRD surveillance, and early recurrence prediction have been extensively reviewed in several studies of HCC patients undergoing LT. Interestingly, some studies showed that TACE can increase ctDNA levels in cell-free DNA in the blood. This might be due to the release of tumor DNA from cancer tissues damaged by TACE. Hence, it is suggested to routinely perform TAE or TACE during diagnostic angiography for HCC to obtain larger amounts of tumor-derived DNA [167].

## 5. Conclusions

Transplant oncology is a promising evolving field in cancer management. The recent push for intense research is creating an extensive optimization of cancer care and patient management. The consolidation of multidisciplinary and collaborative efforts is expected to vastly improve patient outcomes and expedite the expansion of transplant eligibility. Through this measure, we have already seen LT treatment increasingly correlated with improved survival outcomes in patients with liver malignancies. Moreover, the eligibility criteria for LT have expanded beyond the standard Milan criteria over the years to be far more biologically based in order to encompass a wider variety of patients with cancer. In addition, novel techniques, like immunotherapy and ctDNA, are applicable to transplant oncology treatment and are more widely used in recent research studies of oncology in the transplant setting. The options available for immunotherapy use have also been presented as a novel intervention in transplant oncology. Now that it is known that immunotherapy may be used as neoadjuvant “bridging” therapy pre-LT for downstaging and limiting tumor progression, better surgery outcomes are expected. The treatment option of utilizing immunotherapy in the palliative setting post-transplantation has also been studied with promising outcomes for patients. Furthermore, the recent focus of research on liquid biopsy to assess ctDNA post-transplantation can potentially be used as a biomarker to detect MRD and disease recurrence. All of these measures have been comprehensively studied to ensure efficacy and increase survival outcomes in transplant oncology; yet, further investigation is encouraged to establish improved treatment options for cancer patients in the transplant setting.

## Figures and Tables

**Figure 1 cancers-15-05337-f001:**
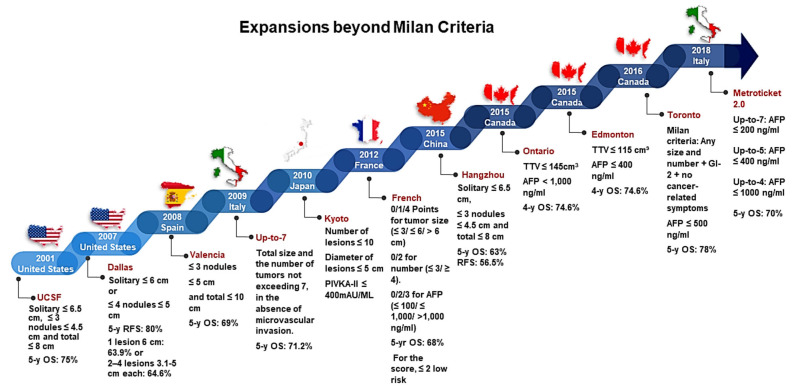
Historical stairway demonstrating progression and expansion of Milan criteria for LT in patients with HCC. **AFP:** alpha fetoprotein, **OS:** overall survival, **RFS:** recurrence-free survival, **TTV:** total tumor volume.

**Figure 2 cancers-15-05337-f002:**
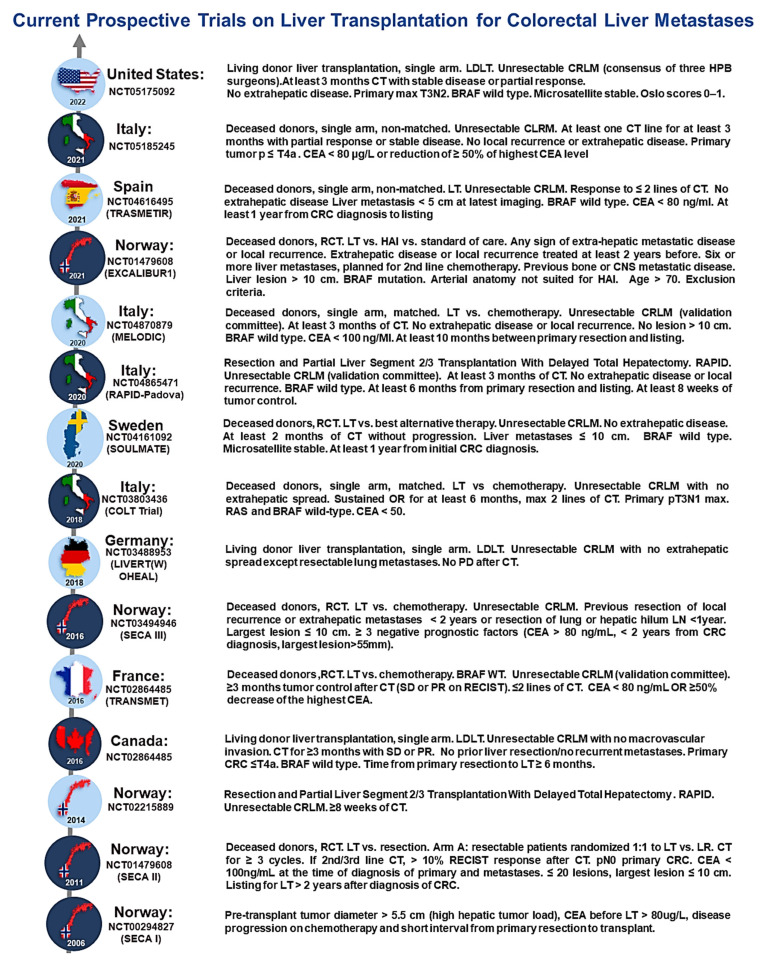
Summary for current prospective trials on LT for colorectal liver metastases. **CRLM:** colorectal liver metastases, **LT:** liver transplant, **TACE:** transarterial chemoembolization, **SIRT:** selective internal radiation therapy, **CT:** chemotherapy, **LDLT**: liver donor liver transplantation, **RPVL:** right portal vein ligation, **LR:** liver resection; **RCT:** randomized controlled trial, **HAI:** hepatic artery infusion, **LDLT**: living donor liver transplantation, **RAPID:** resection and partial liver segment 2–3 transplantation with delayed total hepatectomy, **PD:** progressive disease, **PR**: partial response, **SD**: stable disease, **CEA:** carcinoembryonic antigen.

**Table 1 cancers-15-05337-t001:** Post-LT survival data of most common liver malignancies.

*Malignancy*	*OS (5 y)*	*DFS (5 y)*	*Recurrence*	*References*
HCC	57.7%	65.7%	NA	Li et al. [63]
	71%	72%	NA	Lim et al. [58]
	75% (4 y)	83% (4 y)	NA	Milan Criteria [29]
	75%	NA	NA	UCSF Criteria [38]
	61.8%	80%	NA	Dallas Criteria [39]
	69%	NA	14%	Valencia Criteria [48]
	71.2%	NA	NA	Up-to-7 Criteria [35]
	82%	NA	7%	Kyoto Criteria [33]
	68%	NA	NA	French Criteria [40]
	62.4%	56.5%	NA	Hangzhou Criteria [41]
	74.6% (4 y)	NA	NA	Edmonton Criteria [43]
	78%	NA	NA	Toronto Criteria [46]
	70%	NA	NA	Metroticket 2.0 [45]
HCCA	17%	92%	9%	De Vreede et al. [65]
	29%	33%	NA	Hong et al. [66]
	30%	30%	53%	Robles et al. [67]
ICCA	18%	31%	60%	Casavilla et al. [68]
	23%	NA	51%	Meyer et al. [69]
	21.5%	21.5%	>50%	Panayotova et al. [70]
	65%	18%	NA	Sapisochin et al. [71]
HBL	78%	82%	28%	Pham et al. [72]
NETLM	52%	30%	NA	Le Treut et al. (2013) [73]
	47%	20%	NA	Le Treut et al. (2008) [74]
	48%	32%	NA	Gedaly et al. [75]
	80%	21%	NA	Rosenau et al. [76]
CRLM	60%	NA	90% *	Hagness et al. [77]

**HCC:** hepatocellular carcinoma, **HCCA**: hilar cholangiocarcinoma, **ICCA**: intrahepatic cholangiocarcinoma, **HBL**: hepatoblastoma, **NETLM**: neuroendocrine tumor liver metastasis, **CRLM**: colorectal liver metastasis, **OS**: overall survival, **DFS**: disease-free survival, **LT**: liver transplant, **NA**: not available. * Noted that all known recurrences were easily resectable.

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
