# Peer review of "Transplant Oncology: An Emerging Discipline of Cancer Treatment"

_cancers, 2023, doi:10.3390/cancers15225337_

Round 1
Reviewer 1 Report (Previous Reviewer 2)
Comments and Suggestions for Authors
no additional comments
Reviewer 2 Report (Previous Reviewer 3)
Comments and Suggestions for Authors
The authors addressed the questions. On my site, the manuscript is now suitable for publication.
This manuscript is a resubmission of an earlier submission. The following is a list of the peer review reports and author responses from that submission.
Round 1
Reviewer 1 Report
Comments and Suggestions for Authors
All questions were well responded. Thank you.
Reviewer 2 Report
Comments and Suggestions for Authors
none
Reviewer 3 Report
Comments and Suggestions for Authors
The Authors of this narrative review illustrated a new topic: transplant oncology. However, other reviews have already been published, according to this fact to be published, this review will summarise all the evidence reported in the literature.
Nevertheless, several important steps are described. However, at least two major issues need to be introduced and detailed.
1. The authors reported the results achieved by liver transplantation and compared them with non-transplant alternatives for treating hepatocellular carcinoma. This approach should be implemented for other neoplastic diseases to underline the advantage (if any) of the transplant approach.
2. The Authors reported the results of various experiences in the 4 main paragraphs (Hepatocellular Carcinoma, non-hepatocellular carcinomas (cholangiocarcinoma, hepatoblastoma, liver metastasis from colorectal cancer, liver metastases from neuroendocrine tumours). However, a table summarising the outcome (patient survival, disease-free survival) registered in the largest series will improve the quality of the review.